# Effects of Filtration Processes on the Quality of Extra-Virgin Olive Oil—Literature Update

**DOI:** 10.3390/foods12152918

**Published:** 2023-07-31

**Authors:** Maria Teresa Frangipane, Massimo Cecchini, Danilo Monarca, Riccardo Massantini

**Affiliations:** 1Department for Innovation in Biological, Agro-Food and Forest Systems (DIBAF), University of Tuscia, Via San Camillo de Lellis, 01100 Viterbo, Italy; massanti@unitus.it; 2Department of Agriculture and Forest Sciences (DAFNE), University of Tuscia, Via San Camillo de Lellis, 01100 Viterbo, Italy; cecchini@unitus.it (M.C.); monarca@unitus.it (D.M.); 3Study Alpine Centre, Campus University of Tuscia, Via Rovigo, 738050 Rovigo, Italy

**Keywords:** extra-virgin olive oil, filtration, environmental sustainability, quality

## Abstract

Filtration is a process that eliminates solid sediments and moisture in olive oil to maintain its shelf life during storage. The influence of filtration on the oil characteristics is linked to many parameters such as chemical and sensory traits, cultivars and filtration systems. After assessing the literature on filtration research, we observed that there are contrasting findings and it is complicated to answer the question of whether to filter or not. An analysis of the influence of different filtration technologies used in extra-virgin olive oil production on the phenolic compounds, volatile fractions, antioxidant activity and sensory characteristics is given in this review. The information compiled could help olive oil producers to enhance extra-virgin olive oil quality and maintain it during storage.

## 1. Introduction

Virgin olive oil (VOO) has the characteristic of being an oil produced from fruits and not from seeds. This fact makes it a unique and precious food. Moreover, it is obtained exclusively by mechanical and physical processes such as washing and crushing of olive fruits, malaxation of olive paste, centrifugation and facultative filtration before oil packaging [1]. According to the IOC [2], based on the sensory and chemical analysis, marketable olive oils are classified as: Extra-virgin olive oil (EVOO), Virgin olive oil (VOO) and Ordinary virgin olive oil (OVOO). Of these, extra-virgin olive oil is the grade of the highest quality. Several researchers [3,4,5,6,7,8,9] demonstrated that filtration is crucial as the process’s final step to maintain the qualitative characteristics of the product. During filtering, in fact, important quantitative and qualitative changes take place. The effects of filtration, which removes suspended particles in the olive oil before storage, can positively or negatively influence the quality parameters of the olive oil and its shelf life. In this regard, it has been reported that many factors influence the effects of filtration on the oil, such as the initial chemical composition, the cultivar and the filtration method [8,10,11,12]. In the literature data on filtration research, we have discovered that there is no facile answer to the question of whether to filter olive oil or not since there are reported controversial effects of filtration on the quality of olive oil [5,6,8,10,11,12,13,14,15,16,17,18,19]. It has been amply demonstrated [20] that filtration represents an important stabilization process for extra-virgin olive oil (EVOO) since it is an operation that filters the oil removing suspended solids and enzymes that can bring forward oil decay. Filtration, therefore, remains to this day one of the most debated and controversial phases in the process of producing olive oil. While on the one hand many consumers appreciate the veiled appearance of freshly produced oils and interest in unfiltered oils has increased in the last few years [17,21], on the other hand, over time, unfiltered oils form a deposit on the bottom of the bottle which is not very acceptable. Furthermore, unfiltered oil is more subject to alterations. Indeed, a veiled oil contains water and enzymes which, during storage, can alter the quality of the product. As far as water is concerned, it is for the Olive Council international organization (IOC) to identify the value above which it would be inadvisable to store olive oil with 0.2% of the content [22]. The solid part comes from the olive fragments which remain in suspension after the extraction process. This part contains proteins, polyphenols, phospholipids and glycosides. Chemically these compounds are characterized partly through the lipid phase and partly through the aqueous phase. Because of this, colloidal associations are formed which give the freshly produced oil its typical veiled appearance [23]. Regarding the effect of different filtration systems, it is important to underline that many researchers [3,9,12,18,24,25,26,27] found significant differences both in the shelf life and physicochemical and sensory characteristics of tested olive oils. The filtration systems applied in the olive oil industry can be of different types: there are conventional filtration systems (filter tanks and filter presses), cross-flow filtration (tangential filtration), inert gas-flow filtration systems, and filter bags, as reviewed by Lozano-Sánchez et al. [25]. Most of the studies present in the literature [3,12,18,24,25,26] reported that the diverse types of filtration significantly influence the chemical and sensory quality of extra-virgin olive oils. The present review centers attention on the different filtration technologies used in EVOO production and their influence on quality with a particular emphasis on phenolic compounds, the volatile fraction, antioxidant activity and sensory characteristics.

## 2. Theoretical Aspects of Olive Oil Filtration

Fresh virgin olive oil contains amounts of micro-droplets of vegetation water and solid dispersed particles [28]. This gives a cloudy appearance to the product due to a complex colloid emulsion-sol [29]. Filtration is a separation technique that consists of forcing a suspension through a porous medium capable of retaining the solid part, allowing the liquid component to pass through it, which will then be clarified. Filtration is based on two physical principles: “surface filtration” and “depth filtration”. The first retains the particles, working like a colander, i.e., retaining all the particles with a diameter greater than that of the holes. In depth filtration, in contrast, the solid component is separated from the liquid thanks to the interactions between the filtering medium and the suspended solids themselves, which are retained mainly thanks to adsorption actions, electrostatic attractions, pressure drops and Van der Waals forces. The filtering septum is composed of porous material through which oil passes by gravity and its solid and colloidal particles are retained by absorption within the filter. In this regard, it has been reported that the layer of solid particles deposited on a surface filter exerts an increasingly greater depth action as it increases in thickness [11]. From a theoretical point of view, the speed with which the filtration process takes place, defined as the quantity of filtrate (V) processed per unit of time (*θ*), is controlled by the Hagen Poiseuille equation [30]:

1/A dV/d*θ* = ΔP/µ (αωV/A + r)

A = surface of the filter;

dV/d*θ* = filtration rate (volume of filtrate per unit time);

ΔP = difference of pressures at the two sides of the filter medium;

μ = viscosity;

α = a coefficient due to the filtered resistance that has accumulated in the medium;

w = content of solids in the filtrate in weight per unit of volume;

r = resistance of the filter medium;

From the analysis of the Hagen-Poiseuille equation, it can be observed that an increase in the filter area, A, is correlated to an increase in the filtration speed dV/d*θ*. Therefore, in practice, a very notable role is mainly played by the surface of the filter. As is known, the temperature heavily affects viscosity. As the temperature increases, the viscosity decreases and the filtration rate increases proportionally. Thus, the filtration rate is in inverse proportion to the oil viscosity [31]. This parameter is very important since, in practical applications, the filtration speed of virgin olive oil at 15 °C is half in comparison to that of filtration at 30 °C. Peri [32] highlighted that the optimal temperature for extra-virgin olive oil filtration is 22–24 °C. The parameter α can modify during filtration. During the process, α increases progressively due to the accumulation of solids on the filter. In this case, it was observed that the filtration rate decreased [31]. 

## 3. Filtration Systems of Extra-Virgin Olive Oil

In the olive oil industry, different types of filtration systems can be applied: conventional filtration systems (filter tanks and filter presses), cross-flow filtration (tangential filtration), inert gas-flow filtration systems and filter bags [25]. Conventional systems are founded on the use of organic or inorganic filter aids in conjunction with filtration equipment (tanks or presses) to allow suspended solids and water−oil separations. Among the conventional filtration systems, the more widely used are vertical leaf filter and horizontal filter presses (Figure 1 and Figure 2). Peri [32] published a diagram of filtering equipment. As it is possible to observe, filter sheets are interspersed with filter plates, which can be ‘feeding plates’ (the turbid oil is fed under pressure) or ‘collecting plates’ (the limpid filtrate is gathered and unloaded). Depending on the total amount of oil to be filtered and the desired degree of filtration, it is possible to change the number of filter sheets and plates. Cross-flow filtration is distinguished by a perpendicular flow through the filter and it is often mentioned as Tangential Flow Filtration since its flow is parallel to the membrane. In any case, this technology is not widely used in the olive oil industry due to the physicochemical properties of olive oil [3]. The filter-bag system, developed by Filterflo of Binasco (Milano, Italy), is composed of a cylindrical tube and filter bag [3]. In this technique, olive oil is directly inserted into the polypropylene filter bag from storage tanks. When the oil goes through the filter bag, suspended solids are eliminated. The particularly interesting aspects of this system are its easy maintenance and the excellent degree of oil limpidity. The other filtration system recommended by the University of Bologna and Sapio [33] is founded on the flow of an inert gas (nitrogen or argon).

The characteristic of this method is based on introducing a constant inert gas flow squarely through the olive oil mass during filtration. In this way, it is possible to obtain two advantages: the gas movement makes the precipitation of the suspended solids easy and, at the end of the filtration, the olive oil is already under inert gas that, with the reduction in oxygen, prolongs the shelf life of the product [3].

## 4. Effect of Filtration Technology on Phenolic Compounds

The presence of phenolic compounds in olive oil is ever-topical and a topic of great interest, both for their antioxidant and nutraceutical properties and for the high stability that they provide to olive oil during storage, both of which are sensory characteristics [34,35,36,37]. Several kinds of research [3,5,11,12,16,19,24,25] have been conducted to clarify the relationship between filtration technology and the content of virgin olive oil phenolic compounds (Table 1). The literature on the effect of the filtration process on phenolic compounds is controversial. Some authors reported that phenols increased with filtration [19,25,38]. Guerrini et al. [19] investigated the effect of filtration on virgin oil quality during storage. The filtration process had no significant effect on the total phenolic compound content, but the authors found the profile of the phenolic compounds was significantly modified. After 15 days of storage, the unfiltered samples had a lower content of oleuropein and its derivatives than the filtered ones, with values of 229 vs. 349 mg/kg, respectively. In the same way, the 3,4-DHPEA-EDA content showed a significant increase in the filtered oil samples compared to those unfiltered (123 vs. 67 mg/kg). The authors explain this increase as an effect of the high level of water activity in the unfiltered oil samples since filtration allowed the removal of water as well as solid particles. Indeed, oleuropein and ligstroside transform, resulting in hydrolysis, which produces aglycones (3,4-DHPEA-EA—oleuropein aglycone; p-HPEA-EA—ligstroside aglycone). 

These compounds can generate the corresponding dialdehydic forms which, in turn, decarboxylate into the respective aglycones (3,4-DHPEA-EDA—dialdehydic form of decarboxymethyl oleuropein aglycone; p-HPEA-EDA—dialdehydic form of decarboxymethyl ligstroside aglycone). These results are partially in agreement with those reported by Jabeur et al. [38] who found that secoiridoids are the most influenced phenolic compounds during the filtration process. Their concentrations increase significantly (*p* < 0.05) apart from the decarboxylated ligstroside aglycone, which decreases. Data recorded a significant increase (*p* < 0.05) for the filtered olive oil, mainly of oleuropein aglycone (3,4-DHPEA-EA), with values of 72.30 vs. 36.19 mg/kg in unfiltered samples. In filtered oils, the dialdehydic form of elenolic acid linked to hydroxytyrosol (3,4-DHPEA-EDA) had a content of 13.26 vs. 10.20 mg/kg of the unfiltered sample, such as the ligstroside aglycone (p-HPEA-EA) which showed an increasing amount in filtered oils in comparison to unfiltered ones (28.45 vs. 21.89 mg/kg). The above data have been explained by the authors considering that the removal of water, caused by the filtration process, affected individual phenolic compounds differently as, within the same family, specific chemical characteristics caused different reactions in the filtered oils. Kishimoto [18] compared the processing effects of gravity-driven and mechanical filtration systems on oil quality. He observed a decrease in total phenolic content in oils processed with both gravity-driven and mechanical filtration systems (380 vs. 378 mg/kg) compared to unfiltered oil (430 mg/kg). However, the oxidative stability test demonstrated that both filtered oils had higher stability than unfiltered oil, with a positive impact on the final quality. The author’s findings intimated that filtration should be carried out before storage. Sacchi et al. [13] investigated the content of total phenolic compounds as influenced by the filtering of extra-virgin olive oil from an Italian cultivar, Ravece. Their results reported total phenolic compounds measured both by HPLC and by the Folin–Ciocalteu method were not significantly different for filtered and unfiltered samples. In agreement with other researchers [38], the authors found that 3,4-DHPEA-EA was the most affected phenolic compound; its concentration decreased from 54.46 to 45.68 mg/kg after filtration. These results have been explained by the authors since filtration with water loss is known to decrease the rate of secoiridoid aglycone hydrolysis. The differences found in the amounts of filtered in comparison to unfiltered oils could be attributed to these phenomena. Some researchers [5] evaluated the content of phenolic compounds in extra-virgin olive oil during industrial filtration, which is largely used as a final phase before marketing the oil. Their research confirmed that the influence of filtration on phenolic compounds was different for each family. The authors found phenolic alcohols and flavones decreased after filtration while secoiridoids increased. In regard to secoiridoids, data showed that oleuropein aglycone (3,4-DHPEA-EA), decarboxymethyl oleuropein aglycone (3,4-DHPEA-EDA) and decarboxymethyl ligstroside aglycone (p-HPEA-EA) were the principal compounds notable for their increase after filtration. The authors explained these results by considering how the diverse comportment between each phenolic family after filtration could be linked to their different chemical structures. This influences the division between water fraction and oil arising in a loss of polar phenols in the filtration process, coupled with lowering the water content. The same researchers in another work [41] proposed a correction coefficient able to discern the real comportment of the secoiridoid family in industrial filtration. The use of the correction coefficient confirmed that all the secoiridoid families tended to decrease after filtration. Nevertheless, these data disagree with those reported later by Jabeur et al. [38] who found that secoiridoid concentrations increase significantly during filtration. As a result, the findings show that the influence of filtration on this group of compounds remains unclear. These contradictory findings have been recently confirmed by Ghanbari Shendi et al. [11], who investigated the influence of filtration on the phenols in virgin olive oil extracted from the Turkish cultivar Saurani. Filtration caused a decrease in the content of some phenolic acids, such as 3,4-dihydroxy benzoic acid (0.63 and 0.39 ppm for unfiltered and filtered oils, respectively) and syringic acid (0.17 ppm for unfiltered and not detected for filtered oils, respectively). In filtered oils, the luteolin content was 294.74 ppm vs. unfiltered ones with 347.57 ppm. The same trend was seen in total phenolic content which was higher in unfiltered samples in comparison to filtered oils (565 vs. 520 mg/kg, respectively). From the analysis of the literature, it is possible to observe that numerous authors have found decreases in polyphenols, attributing this to the reduction in the moisture content of the oil during the filtration process. On the other hand, other researchers reported an improvement in shelf life due to the removal of sediments by filtration [16,39]. A similar conclusion was also obtained in a study that investigated the effects of water and solid particles on the stability of olive oil [16]. The authors affirmed that to maintain the quality during storage it is very important to filter produced olive oil quickly. In agreement with this finding are the data of Breschi et al. [39] who characterized olive oil turbidity in order to evaluate the risk of degradation. They found that unfiltered oils were contaminated by microorganisms with a microbial cell count of 6519 (UFC/g), whereas the filtered ones did not show the presence of microorganisms. The authors interpreted these results as linked to the water content and turbidity of the oils. In fact, unfiltered samples showed the highest content of water (0.24% *w*/*w*) and solid particles (0.23% *w*/*w*) and the highest mean turbidity value (1296 NTU, nephelometric turbidity units), whereas the filtered samples showed an almost complete lack of water (0.05% *w*/*w*), no solid particle content and the lowest degree of turbidity (15 NTU, nephelometric turbidity units). Due to the affinity of phenolic compounds for the water phase, the removal of water through a filtration process decreased the total phenolic content by approximately 20%. The total phenolic content of the unfiltered samples was higher than the filtered ones (708 mg vs. 559 mg tyrosol/kg). With the aim of extending the studies on this topic, some researchers [12] observed the effects on the phenolic content of different filtration systems. Oil filtration was conducted using both a cotton filter and a cellulose filter press, on samples from the Coratina and Nera di Colletorto cultivars. The authors reported that the phenolic content modifications were conditioned by the filtration system. As expected, both the filtered monocultivar oils had a lower total phenolic content in comparison to the unfiltered oils. The evidence showed that filtration through a cotton filter decreased the concentration of phenols with values from 679.98 to 589.82 mg/kg in comparison to the unfiltered Coratina cultivar oils. Meanwhile, in the Nera di Colletorto cultivar filtered oils, a reduction in phenols of 7% was noted (330.88 vs. 308.48 mg/kg in unfiltered and filtered samples, respectively). It was interesting to observe that filtration with a cellulose filter press caused a greater reduction in phenolic content in comparison to a cotton filter, by 391.94 and 216.02 mg/kg in the filtered oil of the Coratina and Nera di Colletorto cultivars compared to the unfiltered oil with values of 679.98 vs. 330.88 mg/kg, respectively. The researchers proposed that a cellulose filter could be appropriate for the filtration of unduly cloudy oils. However, filtration with a cotton filter guaranteed better protection of phenolic compounds in the samples, maintaining the nutritional and health properties of the filtered oil. Phenolic fractions in veiled and filtered olive oils produced in the different geographical locations of Tunisia, Spain, Greece and Italy were analyzed by Veneziani et al. [15]. They were found to be reduced in all samples, although the percentage decreases were diverse for Tunisian, Italian, Spanish and Greek filtered oils (54.8%, 34.5%, 28.9% and 28.9%, respectively). Among the phenolic compounds investigated, 3,4-DHPEA-EDA showed a significant decrease: from 37.6 mg/kg in the unfiltered Tunisian samples to 12.1 mg/kg in the filtered ones; 97.5 mg/kg in the unfiltered Spanish samples to 63.8 mg/kg in the filtered ones; 279.2 mg/kg in the unfiltered Greek samples to 163.6 mg/kg in the filtered ones; 187.8 mg/kg in the unfiltered Italian samples to 85.5 mg/kg in the filtered ones. These data, in agreement with those reported by Guerrini et al. [19], confirmed that the losses are mainly due to a reduction in phenyl alcohols and aglyconic derivatives of oleuropein, while the ligstroside derivatives were less influenced by the filtration. Similar results were also obtained by Köseoğlu et al. [40] for three Turkish olive cultivars (Ayvalık, Memecik and Domat). A significant difference between filtered and unfiltered samples was reported for total phenolic content in all the cultivars. They found values of 36.73 mg/kg in the unfiltered Ayvalık cv vs. 28.98 mg/kg in the filtered one; 127.0 mg/kg in the unfiltered Memecik cv vs. 90.95 mg/kg in the filtered one; 104.98 mg/kg in the unfiltered Domat cv vs. 88.02 mg/kg in the filtered one. The authors, in agreement with Jabeur et al. [38], explained these decreases as being because the phenolic compounds, linked to the water droplets present in the olive oil, are eliminated precisely along with the water removed from the filtration. These data are in disagreement with those reported by Vidal et al. [16] who observed the influence of filtration on phenolic compounds in olive oil obtained from the Picual cv. Their findings show no significant differences in the total phenol content of both filtered and unfiltered samples (221.40 and 224.75 mg/kg for filtered and unfiltered, respectively). Despite the discordances observed in the results, the literature analysis leads to the conclusion that filtration is advised because wetness reduction ameliorates the quality of olive oils, due to the higher polar phase content in unfiltered olive oils which may augment the degradation process and reduce the shelf life [42].

## 5. Effect of Filtration Technology on Volatile Compounds

Volatile compounds are a complicated blend of aldehydes, alcohols, ketones, acids, hydrocarbons, and esters and are closely associated with both positive and negative sensory attributes [43]. As described in the literature [44], the primary volatile fractions representative of a pleasant flavor in virgin olive oils are lipoxygenase pathway (LOX pathway) volatile compounds with five (C5) and six (C6) carbon atoms. Guerrini et al. [44] considered two possibilities for filtration: delaying it or filtering it immediately. The aim of their study was to optimize the scheduling of the filtration process, assessing the effects of the filtration of olive oil on volatile compound contents. Findings identified statistically significant differences in the LOX volatile compounds from the C6 and C5 branches related to the filtration process. A significant decrease in the LOX volatile-compound content for filtered samples was observed (Table 2). It is important to underline that the authors found a different reduction related to the chemical properties of diverse volatiles. The decrease was reported as −53.1% for hexanal (0.43 vs 0.20 mg/kg in unfiltered and filtered oils, respectively). Z-3-Hexenal content was 1.05 vs. 0.68 mg/kg in unfiltered and filtered oils, respectively. This evident decrease is coherent with data reported in the literature [6,9,14] and it was explained by researchers as possibly being related to the enzymes involved in the LOX pathway. These enzymes are more active in unfiltered samples due to their residual water content, while in filtered oils they are inhibited by the near absence of water. Moreover, the authors concluded that unfiltered oils spoil more than filtered ones and filtering within a few days reduced the possibility of deteriorated oil quality compared to retarded filtration. These findings agreed with the results of Breschi et al. [17]: the content of C5 and C6 volatile compounds of the LOX pathway was higher in unfiltered samples than in filtered ones. A statistically significant decrease was detected in 1-hexanol, E-2-hexenol, Z-3-hexenol, 1-penten-3-one, and E-2-penten-1-ol volatile compounds, after filtration. This was especially noted in 1-hexanol (0.85 vs. 0.50 mg/kg in unfiltered and filtered samples, respectively), E-2-hexenol (3 vs. 0.00 mg/kg in unfiltered and filtered samples, respectively) and Z-3-hexenol (1.25 vs. 0.78 mg/kg in unfiltered and filtered samples, respectively). It is generally accepted that the water content favors the enzymatic pathway of lipoxygenase, which is responsible for the formation of volatile compounds [44]. However, the authors noted that the unfiltered samples (with a major amount of water content) had a significantly lower level of volatiles in comparison to those filtered. These results confirm those of Brkić Bubola et al. [4] who reported the influence of filtration of Buža and Črna Croatian monovarietal virgin olive oils on volatile compounds. The interesting revelation concerned the fact that the filtration of olive oils affected the volatile compounds differently depending on the cultivar from which the olive oils were obtained. In fact, a significant increase in total alcohols after the filtration of Buža oils was found (4.674 vs. 5.056 mg/kg in unfiltered and filtered samples, respectively). 

Meanwhile, in Črna oils a significant decrease in the totals of alcohols, aldehydes, ketones and C5 volatile compounds was detected. In fact, in Črna filtered-oil samples a significant decrease of 34% in relation to unfiltered ones for total alcohol concentration (8.495 vs. 5.607 mg/kg, respectively) was found. Moreover, in Črna filtered oils, the total amount of aldehydes increased by 9.6% in comparison to unfiltered samples, especially E-2-hexanal with values of 19.962 vs. 21.952 mg/kg, respectively. The filtration process had no significant influence on the total C6 volatiles for both Buža and Črna oils. However, the total of C5 volatile compounds for Črna oils decreased significantly by 4.9% in filtered versus unfiltered oils. The above data were explained by the authors considering that the filtration process had a diverse influence depending on the different varieties of olives. They put forward as possible causes both the complex release mechanism between the oil and water phases and the different amounts of tissue particles removed through filtration. Similar results were also obtained by Fortini et al. [6] who analyzed the effect of filtration on volatile fractions in olive oil samples from Italian olives of the Frantoio cv. They reported that E-2-hexenal, the most abundant volatile compound related to the LOX pathway, increased after filtration (27.193 vs. 36.968 mg/kg for unfiltered and filtered oils, respectively). The researchers affirmed that this achievement could be due to the inhibition of enzyme activity of alcohol dehydrogenase and alcohol acetyltransferase owing to the removal of water during filtration, which protects C6 aldehydes. Another important contribution was provided by Kishimoto [18] with his study which compared the processing effects of gravity-driven and mechanical filtration systems on the volatile component in olive oils. Results showed that the amounts of important volatiles such as C6 molecules were significantly decreased by gravity-driven filtration but not by mechanical filtration. It follows that mechanical-driven filtration is superior in order to remove particles that produce a decay in quality-maintaining volatiles that correspond to the aroma of olive oil. In this regard, some researchers [45] analyzed the influence on olive oil quality of a filtration system using nitrogen or argon flow in contrast to a traditional filter press. They observed that introducing a constant inert gas flow during filtration had an advantage over commercial filtration systems, since the volatiles linked to positive attributes were not altered in inert gas-clarified samples, which manifested a lower water content. In fact, the total of C6 and C5 molecules presented a significant decrease in unfiltered and filtered samples during storage, while they were unvarying in the samples clarified by inert gases. These data disagree with those reported by Vidal et al. [16] who found no major differences between the unfiltered and filtered samples in volatile fractions. The authors observed a significant decrease in filtered oils with respect to the unfiltered ones only for some volatiles such as (E)-2-Hexenal, (Z)-3-hexenol, and (Z)-3-hexenyl acetate (3.13 vs. 3.27 mg/kg, 2.13 vs. 2.33 mg/kg and 2.44 vs. 3.07 mg/kg, respectively). It is important to underline their findings regarding the decrease in six-carbon-atom volatiles, while five-carbon-atom volatile compounds increased after the filtration process. On the contrary, Breschi et al. [39] found no significant differences in the sum of C5 volatile compounds in the unfiltered and filtered samples (9.81 vs. 9.64 mg/kg, respectively). However, they reported that the effect of the filtration process on C6 volatiles was different according to the volatile compound; E-2-hexenol decreased after filtration (7.76 vs. 7.01 mg/kg for unfiltered and filtered samples, respectively), while E-2-hexenal increased after filtration (35.29 vs. 35.60 mg/kg for unfiltered and filtered samples, respectively). These results are similar to those reported in the literature by Brkić Bubola et al. [4]. However, it should be emphasized that the filtration process in olive oil manifested different effects on different volatile chemical compounds due, among other parameters, both to the different filtration methods and to different cultivars from which the oils were obtained. Another consideration was when Kishimoto and Kashiwagi [46] inquired into the effects of filtration on the volatile compounds in olive oils using an electronic nose. Their findings showed that, after filtration, the quantity of C6 volatile compounds significantly decreased in extra-virgin olive oil. This decrease was linked to the number of filtration cycles. To be precise, the decrease in volatiles was significant after the first filtration cycle and continued to reduce as the oils were filtered. Recently, Cecchi et al. [47] examined the volatile compounds of virgin olive oil and highlighted that filtration represents a crucial process to improve the quality and shelf life of the oil. This could depend on the fact that the oils, filtered by reducing the presence of water, enzymes, sugars and other substrates in the sediment, block the development of sensory defects [12]. 

## 6. Impact of Filtration on Sensory Characteristics

Both phenolic and volatile compounds affect the sensory features of virgin olive oils. The influence of the filtration process on these substances is also reflected in the sensory traits (Table 3). This aspect is particularly important for the positive attributes of fruitiness, bitterness and pungency. In this regard, Bendini et al. [26] demonstrated that the filtered samples had significant greater fruity, bitter and pungent intensities (3.5, 4.4 and 4.4, respectively) in comparison to those unfiltered (with values of 1.9, 1.60 and 1.5, respectively). Added to this, the authors reported that for the unfiltered samples the development of off-flavors over time, such as fusty-muddy and winey, were due to the degradation of proteins and sugars present in micro-dispersion into the unfiltered oil. Moreover, they explained that the reason why the filtered oil is less likely to generate sensory defects is since the quicker separation of moisture from the oil induces a reduction in the possible sites of fermentation. These findings are in agreement with those found in the literature [2,17,23,44,45]. 

Valli et al. [45] highlighted after filtration an increase, even if not significant, in positive sensory traits, such as being fruity, bitter and pungent with values of 4.2, 4.2 and 4.4, respectively, for unfiltered oils, and 4.7, 5.5 and 6.6, respectively, in filtered samples. In addition, they found that during storage the decrease in the sensory attributes was slower in filtered oils than in unfiltered ones. This comportment showed that filtration might help to preserve the positive sensory attributes. The most evident influence of filtration was linked to bitter and pungent attributes that remained higher in filtered samples than in the unfiltered oil, corresponding to the minor degradation of secoiridoids during storage. Some researchers [17] underlined a significant effect of the filtration process on positive sensory attributes. Fruity, bitter and pungent intensities in filtered oils had values of 3.40, 3.42 and 4.96, respectively, in comparison to those unfiltered with values of 3.37, 3.30 and 4.53, respectively. They also highlighted a decrease during storage in these sensory attributes, but slower in filtered oils than in unfiltered ones. This trend showed that filtration plays a decisive role in maintaining positive sensory attributes. In this sense, they explained the high fruitiness in filtered oils compared to the unfiltered ones could be dependent on the greater concentration of (E)-2-hexenal (35.29 vs. 35.60 mg/kg for unfiltered and filtered samples, respectively), which is associated with fruity and green notes of olive oil [39]. Other observations were made by Jabeur et al. [38] who reported a decrease in bitter and pungent attributes after oil filtration from the Chemlali cv. These findings, according to the authors, could be due to the hydrophilic phenolic compounds influencing pungency and bitterness. An interesting study was conducted by Fortini et al. [6] on the effects of filter-press filtration on extra-virgin olive oil quality during its shelf life. Olive oil samples obtained from Tuscan olives of the Frantoio cv (Italy) were analyzed. The sensory evaluation demonstrated that the positive fruity attribute remained in filtered samples after three months of storage (4.5 value), while in the unfiltered ones some defects appeared that impeded quantification (0.0 value). The authors found the intensity of bitter and pungent attributes remained high in filtered oils (values of 4.2 and 5.5, respectively). In contrast, unfiltered samples showed the presence of defects: winey (2.3 value), rancid (0.8 value) and fusty (1.8 value). The authors presumed that the enzymatic activities linked to water content in unfiltered oils were the cause, also with microorganisms, of the formation of defects. The removal of water after filtration also explained the presence of a fruity attribute in filtered samples due to enzyme-activity inhibition. All the findings confirmed filtration as an important process to keep the positive sensory traits of the oil.

## 7. Conclusions and Future Perspectives

The literature analysis revealed that the effects of the filtration process on the quality and shelf life of olive oil are contentious. There is a controversy among the researchers since several authors have reported advantages [18,19,25,26,38,45] and others have signaled disadvantages [11,12,13,14,17] regarding the consequences of the filtration of olive oil. Still others found no significant difference between filtered and unfiltered oils [8,10,16]. As pointed out, unfiltered oils keep polyphenols, but they also have hydrolytic and oxidative enzymes, such as lipase and polyphenol oxidase. This aspect is particularly important for the reduction in oxidative stability due to enzymatic reactions. In contrast, filtered oils, by reducing the presence of water, enzymes and other substrates in the sediment, protect polyphenolic compounds from oxidation and block the development of sensory defects. Moreover, the filtered oils preserve greater positive sensory attributes (fruity, bitter and pungent) in comparison to those unfiltered. Therefore, it is possible to conclude that to preserve quality during storage it is very important to filter olive oil quickly. Some best practices for extra-virgin olive oil filtration have emerged from our review such as carrying out the filtration as soon as possible and filtering the oil under inert gas.

Future trends could be to assess the influence of different filtration systems, choosing the most suitable system for the diverse olive cultivars based on their chemical characteristics, such as polyphenolic content. In our opinion, it would be appropriate to conduct further investigations to assess and characterize the chemical and sensory traits of the olive oil from different cultivars, before and after filtration and during storage, using diverse filtration systems. To furnish a clear answer for whether to filter or not, future studies should investigate the use of industrial-scale filtration systems and also evaluate, among the modern methods, the ones with environmental sustainability. Ultimately, the optimization of the operating conditions in the filtration process could be considered by olive oil producers to improve the overall quality level and maintain it during storage. 

## Figures and Tables

**Figure 1 foods-12-02918-f001:**
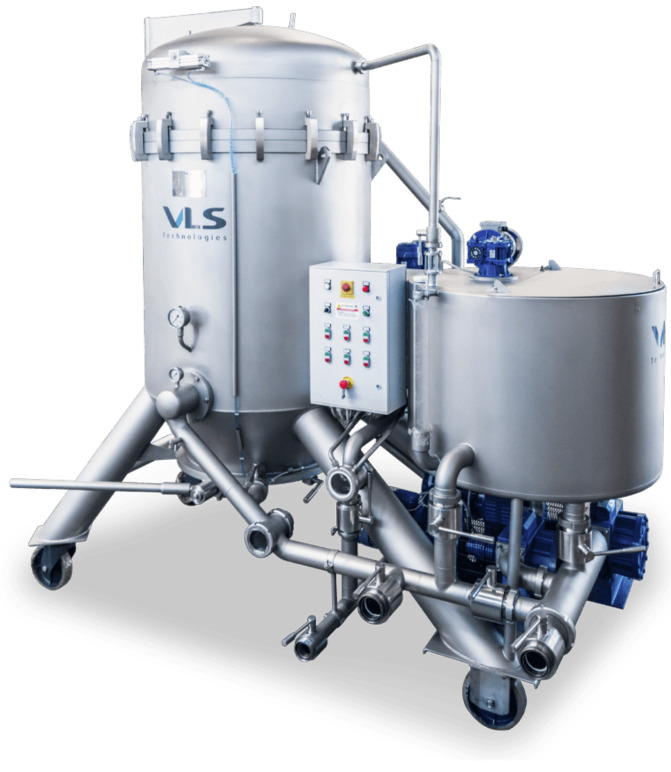
A pressure vertical leaf filter (photos courtesy of VLS Technologies Srl, Ca’Rainati, Italy).

**Figure 2 foods-12-02918-f002:**
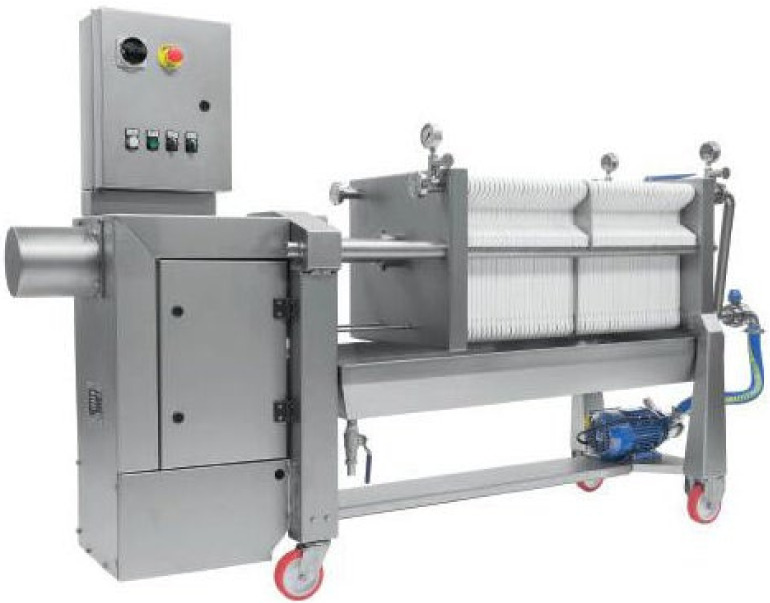
A horizontal filter paper press (photos courtesy of MORI TEM Srl, Sambuca, Italy).

**Table 1 foods-12-02918-t001:** Effect of filtration process on virgin olive oil phenolic compounds (mg/kg).

Cultivars	3,4-DHPEA-EDA		3,4-DHPEA-EA		p-HPEA-EA		Total Plyphenols		References
	U	F	U	F	U	F	U	F	
Correggiolo (Italy)							221	232	Bendini et al. [26]
Frantoio (Italy)	67	123					434	479	Guerrini et al. [19]
Chemlali (Tunisie)	10.20	13.26	36.19	72.30	21.89	28.45	204	198	Jabeur et al. [38]
Olivière (France)							430	380	Kishimoto [18]
Ravece (Italy)	64.52	66	54.46	45.68	8.71	9.59	410.31	431.17	Sacchi et al. [13]
Saurani (Turkey)							565	520	Ghanbari Shendi et al. [11]
Blends of olive cultivars (Italy)	97	88					708	559	Breschi et al. [39]
Coratina (Italy)							679.98	589.82	Zullo et al. [12]
Nera di Colletorto (Italy)							330.88	308.48	
Ayvalık (Turkey)							36.73	28.98	Köseoğlu et al. [40]
Memecik (Turkey)							127	90.95	
Domat (Turkey)							104.98	88.02	
Picual (Spain)							224.75	221.40	Vidal et al. [16]

U: Unfiltered samples; F: Filtered samples.

**Table 2 foods-12-02918-t002:** Effect of filtration process on virgin olive oil volatile compounds (mg/kg).

Cultivars	Hexanal		(E)-2-Hexenal		(E)-2-Pentenal		(Z)-3-Hexenol		(Z)-3-Hexenyl Acetate		E-2-Hexenol		References
	U	F	U	F	U	F	U	F	U	F	U	F	
Blends of cultivars (Italy)	0.43	0.20	170	110	0.20	0.16	0.70	0.60	0.06	0.05			Guerrini et al. [44]
Blends of cultivars (Italy)							1.25	0.78			3	0	Breschi et al. [17]
Buză (Croatia)	0.72	0.74	38.24	38.14			1.42	1.52			2.00	2.14	Brkić Bubola et al. [4]
Črna (Croatia)	0.41	0.38	19.96	21.95			2.29	1.95			3.73	2.53	
Frantoio (Italy)	0.73	0.13	27.19	36.96	0.10	0.12	0.75	0.82	0.06	0.03	6.28	1.011	Fortini et al. [6]
Canino (Italy)	0.67	0.79	14.1	0.79			0.20	0.17			0.40	0.32	Valli et al. [45]
Picual (Spain)	0.56	0.55	3.27	3.13			2.33	2.16	3.07	2.44	0.62	0.65	Vidal et al. [16]
Blends of cultivars (Italy)	0.00	0.00	35.29	35.60			5.24	4.75	0.44	0.40	7.76	7.01	Breschi et al. [39]

U: Unfiltered samples; F: Filtered samples.

**Table 3 foods-12-02918-t003:** Effect of filtration process on virgin olive oil sensory characteristics.

Cultivars	Fruity		Bitter		Pungent		References
	U	F	U	F	U	F	
Correggiolo (Italy)	1.9	3.5	1.6	4.4	1.5	4.4	Bendini et al. [26]
Canino (Italy)	4.2	4.7	4.2	5.5	4.4	6.6	Valli et al. [45]
Blends of cultivars (Italy)	3.37	3.40	3.30	3.42	4.53	4.96	Breschi et al. [17]
Chemlali (Tunisie)	4.2	4	2.4	2.6	2.3	2.5	Jabeur et al. [38]
Coratina (Italy)	6	6.5	4.25	4.5	5.25	5.10	Elsorady et al. [14]
Koronieki (Greece)	5.5	6	3.75	3.65	4.5	4	
Picual (Spain)	5.8	5.6	3.2	2.9	3.9	3.6	Vidal et al. [16]
Beylik (Turkey)	4.4	4	4.4	4.5	5.3	5.5	Ghanbari Shendi et al. [10]

U: Unfiltered samples; F: Filtered samples.

## Data Availability

No new data were created or analyzed in this study. Data sharing is not applicable to this article.

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
