# Peer review of "Effects of Filtration Processes on the Quality of Extra-Virgin Olive Oil—Literature Update"

_foods, 2023, doi:10.3390/foods12152918_

Round 1
Reviewer 1 Report (Previous Reviewer 1)
The changes made by the authors have improved the manuscript.
Reviewer 2 Report (Previous Reviewer 2)
Dear Authors,
I accept this work in present form.
Kind regards
Reviewer 3 Report (Previous Reviewer 3)
...the authors responded to the commends very efficiently. The article is now in amuch better shape and therefore can be published.
This manuscript is a resubmission of an earlier submission. The following is a list of the peer review reports and author responses from that submission.
Round 1
Reviewer 1 Report
The review is written fine, but I am going to suggest some small changes in the text.
Page 1, Line 27: In english, IOOC is written IOC.
Page 2, Line 49: In english, COI is written IOC.
Page 3, line 102: the degree symbol is not correct.
Page 6, table 1: "filtred" is wrong, maybe the word correct is "filtered".
Page 6, table 1: "Filtration" should be in capital letters.
Page 7, line 181 and 184: "filtred" is wrong, maybe the word correct is "filtered".
Page 11, table 2: "the" should be in capital letter.
Page 11, table 2: The part of Vidal (16) is repeated.
The references are not all the same. For example, the doi is not the same in some references as in others. In addition, the doi is missing in some of them.
Out of curiosity, is there anything else on the effect of temperature in literature?
Author Response
We would like to thank the reviewer for providing valuable comments.

Reviewer 2 Report
Dear Authors,
A manuscript titled:
"The influence of the filtration process on the quality of extra virgin olive oil. Literature update" is an interesting and well written review article.
Below are the comments for authors after reading the content of the article. I hope they will help to improve this article.
1. Please change the sentence (line 96-98) “Even if the temperature does not is reported in the equation, some authors [31] found that the temperature heavily affects viscosity.” This sentence misleads readers by suggesting that viscosity only in some cases depends on temperature.
Viscosity strongly depends on temperature! (https://en.wikipedia.org/wiki/Temperature_dependence_of_viscosity)
2. I would ask the authors to make sure that they explain all the abbreviations used in the paper. For example, line 27, What is IOOC?
Sincerely yours,
Reviewer
Author Response

(The authors gave the same response as above.)

Reviewer 3 Report
Effects of Filtration process on the quality of extra virgin olive 2 oil. Literature update
The manuscript is well organised and well written. My commends for the imrovement of the manuscript are as follows:
The title is very general. The term quality includes several other parameters the the phenolic content and profile, the volatiles and the organoleptic attributes.
The tables need further elaboration. As they are, they are but a repetition of the discussion in the text. There is a need for further breakdown of information so that the reader at one glance to combrehent the issue. A third table is needed for the organoleptic traits.
Once the tables are imroved, the discussion on the related chapters will become easier to read.
On the 'Effect of filtration technology on volatile compounds the discussion is rather facilitated by the introduction in the discussion of the enzyme background as a cause of the related changes.
This is not the case, however, for the 'Effect of filtration technology on phenolic compounds' where the differences in specific secoiridoids, could be further explained in terms of glycosidases or other enzymes involved in this pathway. Whether or not these enzymes convert oleuropain/ligstroside and coresponding derivatives to aglycons (during filtration) which in turn possess better affinity with the oil. The elaboration of the table may lead to a hypothesis instead of leaving the issue unresolved.
Further, attention should also be given to the extraction protocols of the phenolic fraction, since the involvement of water may play a role (before and after filtration). Different protocols use different solvents with different affinoty to water/oil and to phenolics (glycosylated/aglycons) in addition to the use (or not) of supersonics.
The sensory characteristics indicate increased phenolics after filtration. Perhaps the article should start with this chapter and then the explanation chapters (on phenolics and volatiles) should follow.
These are my thoughts reading this review. In any case, the two (plus a third on sensory-to be introduced) tables should be reconstructed to be more readable.
Author Response

(The authors gave the same response as above.)

Round 2
Reviewer 3 Report
A table on sensory is added, however, I still find that in the new version, the tables are not elaborated enough to facilitate the reader in comprehending the primary data. I do not wish to make suggestions since this is the authors' responsibility. There is a standard way of presenting such tables in most reviews.
Author Response

(The authors gave the same response as above.)
